# Testing the causal mechanism of the peninsular effect in passerine birds from South Korea

**Jin-Yong Kim**[1], **Man-Seok Shin**[2], **Changwan Seo**[3], **Soo Hyung Eo**[4], **Seungbum Hong**[5]*

**1** Division of Restoration Research, Research Center for Endangered Species, Yeongyang, South Korea, **2** Division of Ecological Information, National Institute of Ecology, Seocheon, South Korea, **3** Division of Ecological Assessment, National Institute of Ecology, Seocheon, South Korea, **4** Department of Forest Resources, Kongju National University, Choongnam, South Korea, **5** Division of Climate & Ecology Research, National Institute of Ecology, Seocheon, South Korea

* sbhong@nie.re.kr

**Data Availability Statement:** NES has been conducted by The South Korean Ministry of Environment since 1986 and the data are available from the EcoBank (website: https://nie-ecobank.kr/rsrch/doi/selectDoiRsrchDtaListVw.do).

## Abstract

The peninsular effect is a geographical phenomenon that explains patterns of species richness. Given that spatial variation in species richness along a peninsular may be driven by multiple processes, we aimed to identify the sources of latitudinal patterns in passerine species richness and test hypotheses regarding (1) recent deterministic processes (climate, primary productivity, forest area, and habitat diversity), (2) anthropogenic processes (habitat fragmentation), and (3) stochastic processes (migration influence) in the Korean peninsula. We used the distribution data of 147 passerine species from 2006 to 2012. Single regression between passerine species richness and latitude supported the existence of the peninsular effect. Mean temperature increased with decreasing latitude, as did habitat diversity but leaf area index and forest area decline. However, mean temperature and forest area only influenced passerine species richness. Although habitat diversity influenced passerine species richness, it was counter to the expectations associated with peninsular effect. The number of habitat patches decreased as latitude increased but it had no effect on passerine species richness. Ratio of migrant species richness showed no significant relationship with leaf area index, forest area, and habitat diversity. However, the ratio of migrant species richness increased with decreasing mean temperature and that contributed to the increase in passerine species. Overall, our finding indicate that the observed species richness pattern in peninsulas with the tip pointing south (in the northern hemisphere) counter to the global latitudinal gradient. These results were caused by the peninsular effect associated with complex mechanism that interact with climate, habitat area, and migrant species inflow.

## Introduction

Gradients of species richness remain some of the only information available to effectively evaluate patterns of biodiversity at broad spatial extents [1] and are one of the Earth's salient biological characteristics [2]. Since the 19th century, the proximate and ultimate causes of species

**Funding:** This study was funded by the Assessment of Climate Change Risk for Ecosystems in Korea (NIE-BR-2020-11) and the Study of adaptation capacity to climate change risk for ecosystem(NIE-BR-2011-35) from the National Institute of Ecology and supported by the Climate Change Response Technology Project of the Ministry of Environment, Republic of Korea (2014001310009).

**Competing interests:** The authors have declared that no competing interests exist.

richness gradients continue to stimulate scientific debate and drive many hypotheses testing in macroecology and biogeography [2]. The peninsular effect [3] is one of such geographical hypothesis that attempts to explain patterns of species richness, and postulates that the number of species inhabiting an area declines from a peninsula's base to its tip. Many studies have been conducted using different taxa at various spatial scales and have proposed a variety of functionally hypotheses describing possible causes of this ecological phenomena [4, 5]. From these previous works, the potential causal mechanisms of the peninsular effect may be grouped as follows: (1) recent stochastic processes (equilibrium and derived island biogeography theories); (2) historical events (paleoclimatic and paleogeographic changes); (3) recent deterministic processes with a geographical base–tip gradient; and (4) recent anthropogenic processes [4].

Recent stochastic processes involve immigration–extinction dynamics, area effect, and isolation effect [6] and together, propose that unsuitable habitats and isolating conditions for specific taxa may be unevenly distributed throughout a peninsula and create certain patterns of species richness [3, 7]. Whereas, the historical events hypothesis proposes that current species distributions are predominantly a result of past climatic or geological events (e.g. glacial retreat) that has determined the rate of species turnover, and shaped patterns in genetic diversity, speciation, and genetic differentiation [6]. In contrast, recent deterministic processes are related to ongoing patterns of habitat heterogeneity and patchiness, climatic regime, vegetation structure, and habitat area [6, 8]. Lastly, recent anthropogenic processes are linked to the gradient of human-driven disturbances, such as deforestation, fires, pasture, and fragmentation which might increase or decrease of species richness due to an increase of anthropophilous generalist species [6, 9].

From an interspecific lens, early work [3] proposed that non-flying taxa (e.g., mammals, amphibians, and plants) would be more constrained by geometric features than flying taxa, and thus show a stronger peninsular effect. However, Lawlor [10] pointed out that this phenomenon was observed only in heteromyd rodents from the Baja California peninsula. However, a recent study proposed that the peninsular effect is more prevalent in flying taxa than in non-flying taxa [5], and more likely to be attributed to the influence of recent deterministic processes such as climate, habitat and anthropogenic factor rather than to historical processes or immigration–extinction dynamics [6]. However, the distribution of bird species throughout a peninsular may be additionally impacted by their, relative to non-flying taxa, greater ability to disperse over water, which may exceed their ability to disperse over land [6, 11]. Therefore, the observed peninsular patterns could be changed by temporal influx of migrant birds and their subsequent interaction with resident birds. Thus, in the case of bird taxa, the temporal influx of migrant competition might be the major drivers for peninsular effect.

In north-south-oriented peninsulas with large climatic range, variation in temperature along a latitudinal gradient and subsequent shifts in vegetation traits (e.g., primary productivity, habitat diversity, and habitat area) is likely the most important determinant of distributional patterns of species richness [6, 12–14]. In general, an increase of bird species richness can be expected when seasonal temperatures decreases and primary productivity, habitat diversity, and habitat area increase [6, 8, 13]. However, if human settlement or land use also varies along a base-tip gradient, human-induced habitat heterogeneity and fragmentation as well as anthropogenic perturbations may further influence the species richness along the same latitudinal gradient described under the peninsula effect [6]. Meanwhile, Herrera [15] discovered that the species richness of migratory passerine increased with increasing latitude in Europe, which was attributed to resident species having a narrower habitat niche than migrant species [16, 17]. Thus, to understand the more specific causal mechanism in bird taxa, it is

important to identify not only environmental changes but temporal changes in migrant influx caused by competition and habitat changes according to the peninsular effect.

This study aimed to test the peninsular effect on passerine birds in South Korea based on the various aforementioned hypotheses. First, we identified the latitudinal pattern of passerine species richness in South Korea. Then, we tested the hypotheses of the peninsular effect related to (1) recent deterministic processes (climate, primary productivity, habitat diversity, and forest area), (2) anthropogenic processes (habitat fragmentation), and (3) stochastic processes (migration influence). For testing these hypotheses, we constructed the conceptual scheme of this study and tested hypotheses according to recent deterministic, anthropogenic, and stochastic processes (Fig 1((a), 1(b)), 1(c), 1((d) and 1(e))), respectively.

## Methods

### Bird data

The Korean peninsula is connected to the Asian continent to north, and its tip is towards the south. Because of the political situation in the Korean peninsula, the present study was limited to South Korea (34–39°N and 126–130°E) over an area of approximately 95,219 km². Distribution data of passerine species from 2006 to 2012 were extracted from the National Ecosystem Survey (NES) in South Korea. NES has been conducted by The South Korean Ministry of Environment since 1986 and the data are available from the EcoBank (website: https://nie-ecobank.kr/rsrch/doi/selectDoiRsrchDtaListVw.do).

The NES survey was conducted in a grid of 0.041 square decimal degrees (17.3km²) within a 0.125 × 0.125 square decimal degree (Fig 2). The survey points (0.041 square decimal degrees) were randomly selected considering a representative mountain area and accessibility in each grid cell (Fig 2). Within the survey point, a linear path way was walked to count bird data [18]. These transect surveys were conducted three times a year between February and November. Species name, number of birds detected for each species, and the geographic location of each species (latitude and longitude) was collected. All data from islands were eliminated from the analysis. Because these islands are used for migratory birds as stopover sites, so species richness might be extremely higher compared with the inland area during migration season [19]. All birds were classified as resident or migrant (S1 Table). Resident denote a species that spends all season in Korean peninsula and migrant denote a species that stop for a while (winter visitor and passage migrant) or visit for the purpose of breeding (summer visitor).

### Climate (temperature) and primary productivity (LAI)

To identify the effect of climate and primary productivity on latitudinal distribution, surface temperature (TSK) and leaf area index (LAI) were obtained from satellite observation data from the Moderate Resolution Imaging Spectro-radiometer (MODIS). MODIS is operated on two satellite platforms, Terra and Aqua, by the United States National Aeronautics and Space Administration (NASA). TSK was retrieved by a generalized split-window algorithm with thermal infrared detections, and it represents temperature at the soil or canopy surface [20]. The TSK data was monthly-averaged gridded data with a 0.05 decimal degree size for each grid. To extract mean temperature, we averaged the TSK data from 2006 to 2013. LAI was defined as the area of a one-sided broad leaf or projected needle leaf per unit ground area. LAI is generally used for quantifying the amount of photosynthetically active leaves. Thus, LAI is considered a key driver of forest productivity [21]. These data are derived from MODIS surface reflectance with structural and optical characteristics of vegetation types [22]. The LAI data herein used were 16-day-composed gridded data in 250 m for each grid. The gridded

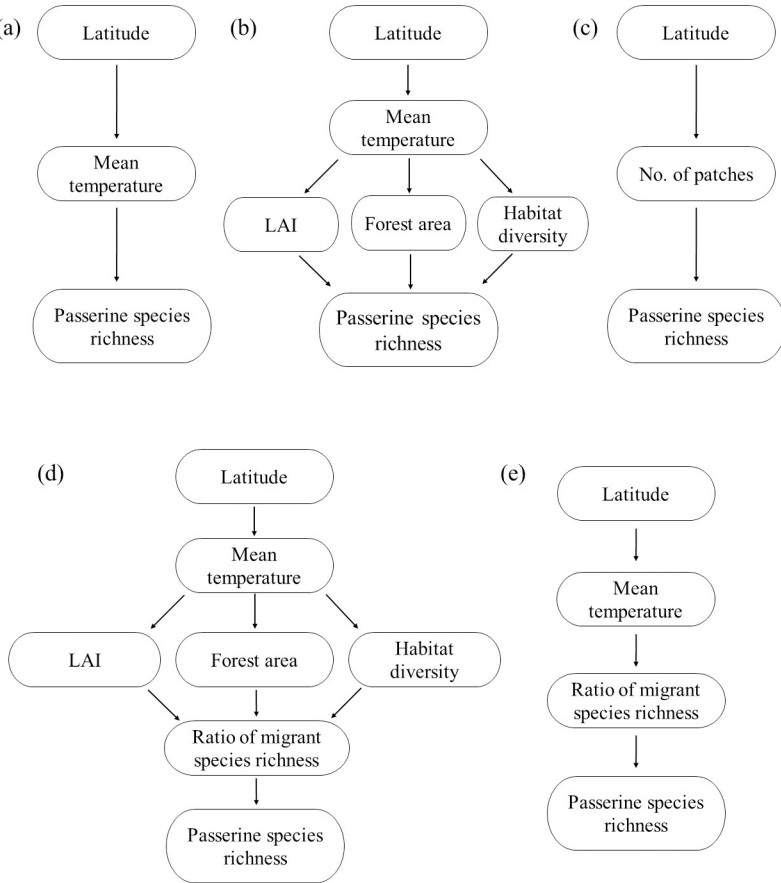

**Fig 1. Conceptual scheme of the study to test the peninsular effect based on recent deterministic ((a) and (b)), anthropogenic (c), and stochastic ((d) and (e)) processes.**

MODIS TSK and LAI data can be obtained via lpdaac.usgs.gov, and all data were averaged across years from 2006 to 2013, which is the same period of the NES data.

## Habitat diversity

A MODIS land cover dataset, which follows the land cover classification from the International Geosphere-Biosphere Programme (IGBP) [23], was used to quantify habitat diversity. The IGBP classification categorizes land covers globally into 17 vegetation types (Fig 3). Among them, land covers in South Korea are classified into 13 IGBP types. Habitat diversity indices were derived from a species diversity index [24, 25]. Thus, habitat diversity in a given unit area was defined by the following equation, which is the same as the typical species richness estimation:

$$D = \sum_{i=1}^{n} \{-A_i/A_{tot} \cdot \ln(A_i/A_{tot})\}, \tag{1}$$

where $A_i$ is the covering area of the land cover type $i$, and $A_{tot}$ is the total area for all land cover types of the given analytical unit. The unit area was set at a $0.125 \times 0.125$ square decimal degrees, the same unit size of the NES data.

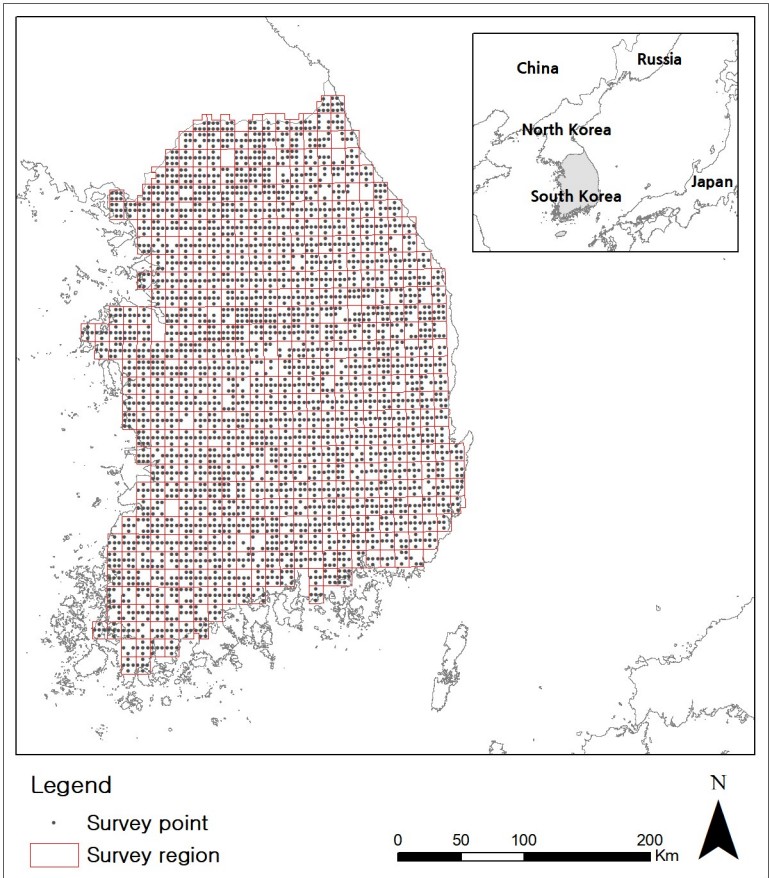

**Fig 2. Location of the Korean peninsula and NES survey regions.**

## Forest area

The data of forest area were extracted from a 1:25,000 scale land cover map from 2009 from the Korea Ministry of Environment (KME, http://egis.me.go.kr). The KME land cover map was produced with a 5 m spatial resolution using satellite imagery (Landsat TM, IRS-1C, SPOT-5, KOMPSAT-2), and it consisted of 22 land cover classes (vegetation or anthropogenic area). Among them, the forest area included in a grid of 0.125 × 0.125 square decimal degree was calculated using the 'calculate geometry' function of ArcGIS in the land cover map. Then, a grid of 0.125 × 0.125 square decimal degree was generated using the fishnet tool in ArcGIS. Spatial data were compiled using the ArcGIS 10.3 and R Studio 1.1.383 software programs.

## Habitat fragmentation (number of patches)

To identify habitat fragmentation induced by anthropogenic disturbances, we used number of patches (No. of patches). Data on forest patches were extracted from the KME land cover map. An identification number (ID) was assigned to each forest patch and grid. Then, using the union tool in ArcGIS, forest patches were overlapped with grid cells. The number of forest patches was obtained by counting the ID numbers of the forest patches included in a grid of 0.125 × 0.125 square decimal degree. The number of forest patches was counted using the dplyr package for R Studio 1.1.383.

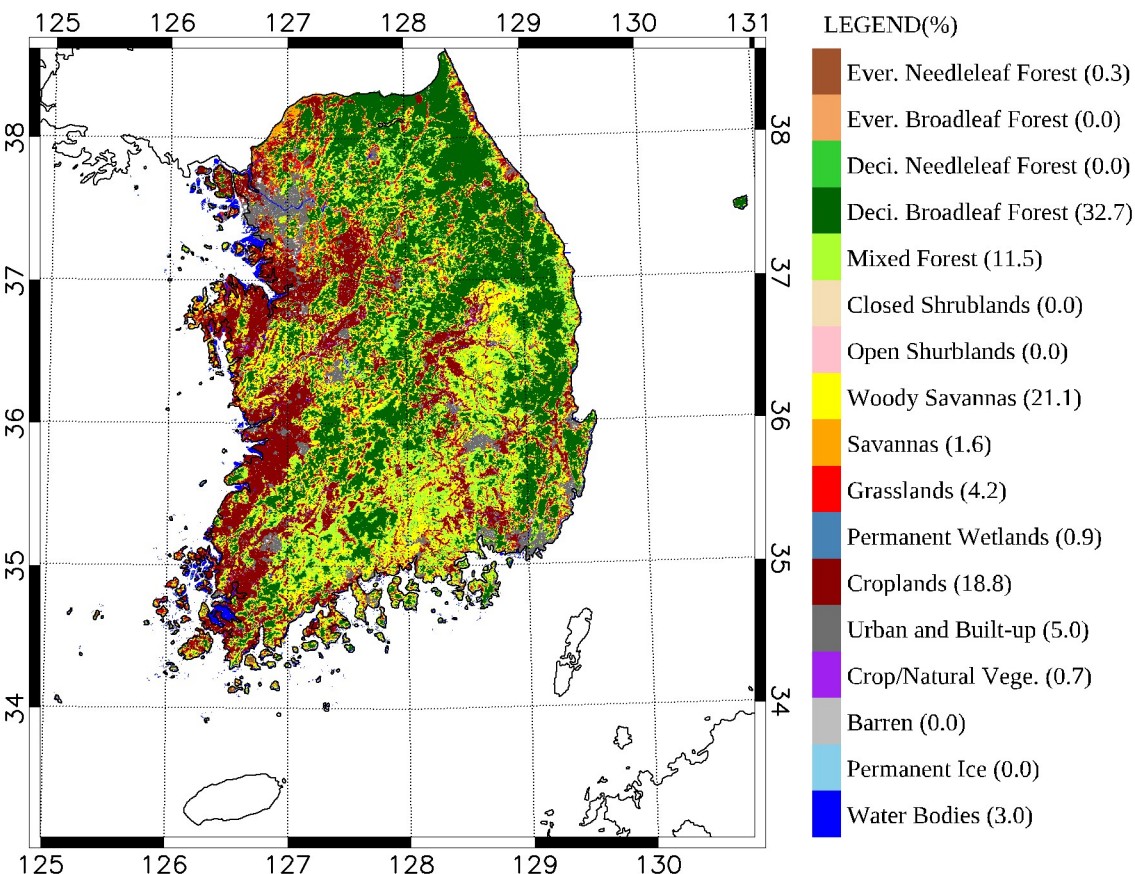

**Fig 3. Moderate Resolution Imaging Spectro-radiometer (MODIS) land cover map for South Korea with the 17 IGBP classification.** The numbers beside the names of land cover types are their percentages to the total land area of South Korea.

## Data analyses

To identify the peninsular effect on passerine birds, patterns of linear regression between passerine species richness and latitude were verified within a grid of $0.125 \times 0.125$ square decimal degree. Species richness was averaged according to resolution scales ($0.125 \times 0.125$ square decimal degree), to reduce differences in sampling effort [6]. Statistical analysis of linear regression was performed using SigmaPlot 13.0 (Systat Software, Inc.) and SPSS statistics 20 (IBM Corp.).

We used piecewise structural equation modeling (pSEM) with a generalized least squares model (GLS) for testing our conceptual models. Because it allow us to account for hierarchy of effects and to investigate the relationship between multiple response and predictor variable [26]. Five pSEMs for testing the conceptual models (Fig 1(a), 1(b), 1(c), 1(d), and 1(e)) were constructed based on recent deterministic and anthropogenic hypotheses. In brief, our conceptual model assumed that 1) latitude would directly affect both mean temperature and No. of patches; 2) temperature would influence each vegetation variables, as well as the ratio of migrant species richness; 3) temperature, LAI, forest area, habitat diversity, ratio of migrant species richness, and No. of patches would ultimately influence passerine species richness. And we considered spatial autocorrelation as a function of a random effect based on coordination of each location [27]. We assessed the models (pSEM) fit to the data using Fisher's C

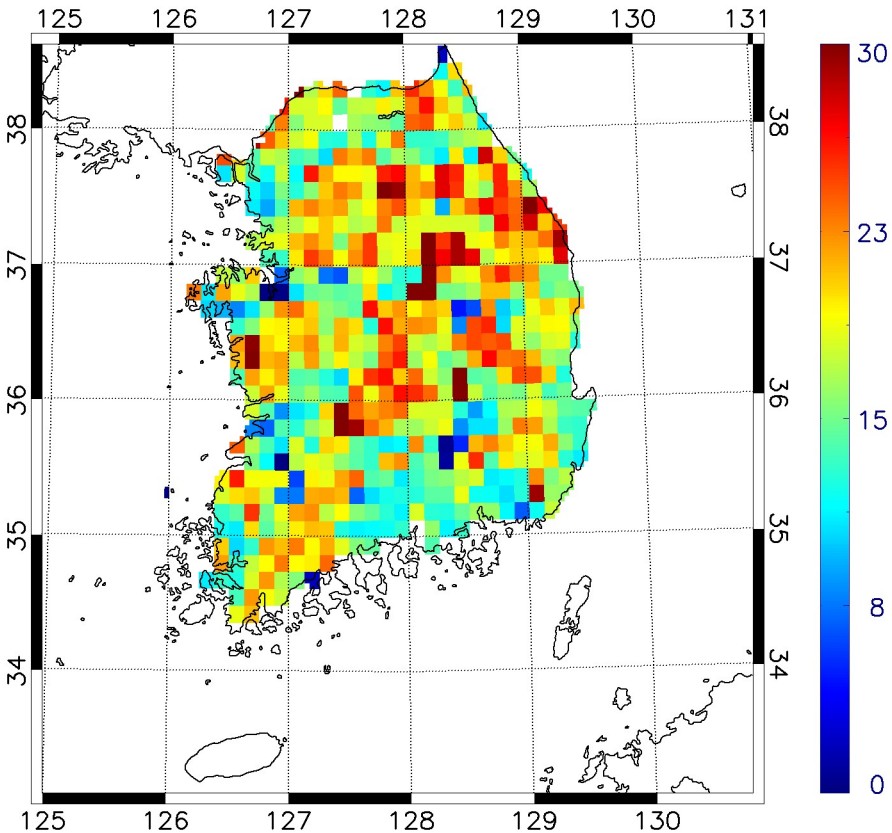

**Fig 4. Distribution of passerine species richness in South Korea.**

statistics and associated *P*-value (i.e. *P* > 0.05 indicates accepted model) [28]. All statistical analyses were performed using R 4.0.0 (packages piecewiseSEM, nlme, lme4).

## Results

### Testing the peninsular effect

A total of 147 passerine species were observed in the 589 regions in lattices of $0.125 \times 0.125°$ resolution (S1 Table). We tested the peninsular effect on 147 passerine species. The linear pattern of single model between passerine species richness and latitude showed that passerine species richness increased with increasing latitude (coefficient: 0.991, Fig 4). The linear pattern represented slightly low $R^2$ than quadratic pattern but a significant increase ($R^2 = 0.036$, $P < 0.001$, S2 Table).

### Testing hypotheses that explain the peninsular effect

In the recent deterministic model suite (S3 Table), the pSEMs supported that higher latitudes were associated with lower mean temperatures ($R^2 = 0.44$, Fig 5(a) and 5(b)), and mean temperature was negatively correlated with passerine species richness directly ($R^2 = 0.08$, Fig 5(a)). Furthermore, mean temperature had a significant negative effect on LAI and forest area ($R^2 = 0.50$ and $R^2 = 0.64$, respectively, Fig 5(b)) but a positive effect on habitat diversity ($R^2 = 0.34$, Fig 5(b)). Both forest area and habitat diversity positively associated with passerine species richness; however, LAI was not associated with passerine species richness ($R^2 = 0.08$, Fig 5(b)).

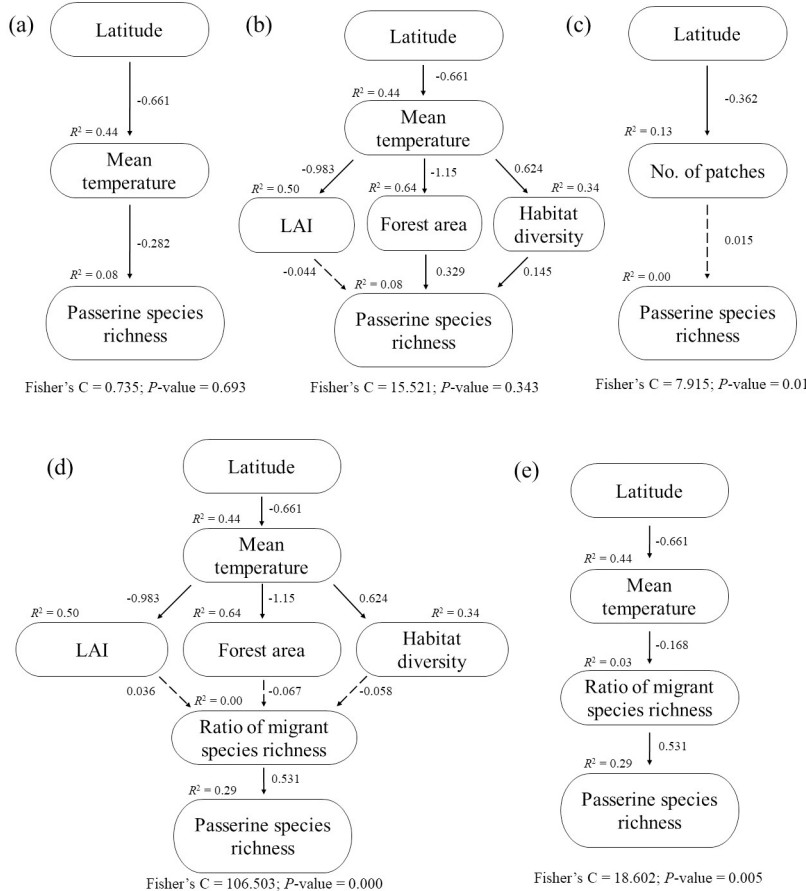

**Fig 5. The key piecewise structural equation models (pSEM) for testing the hypotheses of peninsular effect based on recent deterministic ((a) and (b)), anthropogenic (c), and stochastic ((d) and (e)) processes.** Solid arrows represent significant paths ($P < 0.05$) while dashed arrows represent nonsignificant paths ($P > 0.05$). For each variable, $R^2$ is provided. Model-fit statistics (Fisher's C and $P$-value) for each pSEM are given.

Both climate (Fig 5(a)) and habitat (Fig 5(b)) models were identified as accepted models (Fisher's C: 0.735; $P = 0.693$, Fisher's C: 15.521; $P = 0.343$, respectively).

In anthropogenic model (S3 Table), No. of patches decreased as latitude increased ($R^2 = 0.13$, Fig 5(c)). However, there was no relationship between No. of patches and passerine species richness ($R^2 = 0.00$, Fig 5(c)).

In stochastic models (S3 Table), ratio of migratory species richness showed no significant relationship with LAI, forest area, and habitat diversity ($R^2 = 0.00$, Fig 5(d)). However, mean temperature had a significant negative effect on ratio of migratory species richness ($R^2 = 0.03$, Fig 5(e)). In addition, passerine species richness was increased as ratio of migratory species richness increased ($R^2 = 0.29$, Fig 5(d) and 5(e)). The model-fit statistics of both Fig 5(d) and 5(e) models was not significant (Fisher's C: 106.503; $P = 0.000$, Fisher's C: 18.602; $P = 0.005$, respectively).

## Discussion

### Testing the peninsular effect

Our results of the single regression between passerine species richness and latitude support the existence of the peninsular effect (S2 Table). However, distribution pattern of passerine species

richness showed a decreasing around at ~38˚N (Fig 4 and S2 Table). We assumed that these effects were caused by the decrease in species richness near the northern border; as many army units and troops are deployed near the Military Demarcation Line, conducting surveys on living organisms in that area is difficult and subject to many restrictions. Moreover, the bird survey data used herein were restricted to South Korea for political reasons; thus, we were unable to consider the richness pattern of the entire peninsula. Nevertheless, the richness decrease from the base towards the tip is shown herein. Considering that our results showed the importance of mean temperature, forest area, and migrant inflow on passerine species richness (Fig 5), we expect passerine species richness to increase in North Korea, especially as it is expected containing wider forest areas, higher migrant inflow, and lower mean temperature.

## Testing hypotheses to explain the peninsular effect

Increasing latitude led to decreasing mean temperature, which led to increasing passerine species richness (Fig 5(a)). And forest area, induced by temperature gradient, affected passerine species richness positively (Fig 5(b)). These results support the existence of the peninsular effect on passerine birds to be at least partially induced by climate, habitat area. Our results mirror previous studies conducted in Baja California (birds) and Iberian peninsula (passerine birds) that also demonstrated that species richness on a peninsular varied along a temperature gradient [6, 29, 30]. However, in peninsulas with the tip pointing south, the observed decreasing base-tip diversity patterns could not be coherent with the global latitudinal gradient [6], and against the physiological tolerance hypothesis [31]. Thus, the underlying hypotheses describing the peninsular effect should not only explain the relationship between temperature and species richness. In the global latitudinal gradient generally, species richness of trees decrease monotonically with latitude, and increase with temperature [32]. However, according to the peninsular effect, the lack of the available interior habitat, as well as the progressive increase in edge area towards the tip of the peninsular, are the major causal factors explaining the species richness and occurrence [6, 33].

Previous studies have suggested that habitat diversity might be an important factor explaining the peninsular effect [30, 34]. However, habitat diversity, itself, is a partially function of climatic and edaphic processes, induced by topographic heterogeneity (latitude and altitude) [35, 36]. In our results, habitat diversity increased with increasing mean temperature, whereas LAI and forest area increased with decreasing mean temperature (Fig 5(b)), which is presumed to be due to habitat simplification by increased mature forest and forest area towards the base of the peninsula. These results show primary productivity and forest area are incompatible with habitat diversity. However, habitat diversity had a positive relationship with passerine species richness (Fig 5(b)). These results do not explain the hypothesis of peninsular effect, although habitat diversity and passerine species richness had a significant relationship. Because habitat diversity and passerine species richness must show a negative result to consist with the hypotheses of peninsular effect. In order to extract more accurate habitat diversity for passerine birds, it would be appropriate to use the methods of identifying vegetation structure in forest areas as well as using satellite images.

Our results showed that LAI did not significantly affect passerine birds (Fig 5(a)). The amount of energy available, measured using primary productivity as a proxy, has been considered one of the major determinants of species richness [37]. As birds need food resources the most during the breeding season for rearing chicks, primary productivity may play an important role in the distribution of birds particularly during this season. However, we used bird data from all seasons (rather than from a specific season) in our analysis, which may explain our results not showing a significant effect of LAI on the distribution of passerine birds.

Battisti [6] argued that if the anthropogenic factors have a base-tip gradient, that can influence the peninsular species richness patterns. Our results show that the number of patches decreased with increasing latitude, but passerine species richness was not influenced by habitat patchiness (Fig 5(c)). Recent studies have put forward controversial views regarding the effect of the anthropogenic factors, and have demonstrated that counterintuitive may occur due to anthropophilous generalist species and previous extinctions of sensitive species [6, 38–40]. According to the viewpoint of fragmentation having positive effects for biodiversity, fragmentation is not frequently linked to habitat loss; in contrast, it is advantageous for generalist or invasive species associated with edges for habitat diversity, and it reduces competition [39]. Thus, our results did not show a significant relationship between the number of patches and passerine species richness.

In general, vegetation structure and primary productivity are considered an important variable for most passerine bird species [37, 41] as these species inhabit forest areas to forage, rest, and breed regardless of being resident or migrant. However, there was no significant relationship between ratio of migrant species richness and LAI, forest area, habitat diversity (Fig 5(d)). However, we found that the ratio of migrant species richness increased with decreasing mean temperature and that contributed to the increase in passerine species (Fig 5(e)). The physiological tolerance hypothesis predicts that more combinations of life history strategies can persist in warm and wet than cold or dry conditions [31]. And migrant species have the characteristics of generalist due to competition with resident. Thus, it is assumed that distribution of migratory birds in areas with lower temperature contributed to the increase in passerine species richness. Therefore, these results indicate that migrant species inflow increased passerine species richness and the peninsular effect.

## Conclusion

The single model of passerine species richness showed species richness increased with latitude in South Korea. Mean temperature increased with decreasing latitude, as did habitat diversity but leaf area index and forest area decline. Both forest area and habitat diversity positively associated with passerine species richness; however, leaf area index was not associated with passerine species richness. Although habitat diversity influenced passerine species richness, it was counter to the expectations associated with the peninsular effect. Therefore, we have concluded that only mean temperature and forest area influenced passerine species richness. According to these results, passerines species richness is expected to increase in North Korea. The number of patches decreased as latitude increased but it had no effect on passerine species richness. Ratio of migrant species richness showed no significant relationship with leaf area index, forest area, and habitat diversity. However, ratio of migrant species richness increased with decreasing mean temperature and that contributed to the increase in passerine species. Overall, our findings indicate that the observed decreasing base-tip diversity patterns in peninsulas with the tip pointing south (in the northern hemisphere) counter to the global latitudinal gradient. And these results were caused by the peninsular effect associated with complex mechanism that interact with climate, habitat area, and migrant species inflow.

## Supporting information

**S1 Table. Species checklists of passerine birds recorded in South Korea.**
(PDF)

**S2 Table. Results of linear and quadratic regression between passerine species richness and latitude.**
(PDF)

**S3 Table. The summary of piecewise structural equation modelling (pSEM) for testing the hypotheses of peninsular effect based on recent deterministic, anthropogenic, and stochastic processes.** Significant paths ($P<0.05$) are indicated in bold.
(PDF)

**S1 Data.**
(XLSX)

## Author Contributions

**Conceptualization:** Jin-Yong Kim, Changwan Seo, Soo Hyung Eo, Seungbum Hong.

**Data curation:** Jin-Yong Kim, Man-Seok Shin, Seungbum Hong.

**Formal analysis:** Jin-Yong Kim, Man-Seok Shin, Seungbum Hong.

**Funding acquisition:** Changwan Seo.

**Methodology:** Jin-Yong Kim, Man-Seok Shin, Changwan Seo, Soo Hyung Eo, Seungbum Hong.

**Supervision:** Changwan Seo, Soo Hyung Eo, Seungbum Hong.

**Validation:** Jin-Yong Kim, Man-Seok Shin, Changwan Seo, Soo Hyung Eo, Seungbum Hong.

**Writing – original draft:** Jin-Yong Kim.

**Writing – review & editing:** Jin-Yong Kim, Man-Seok Shin, Changwan Seo, Seungbum Hong.

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
