## [Decision Letter · Decision Letter 0]

15 Dec 2020

PONE-D-20-29245

Testing the causal mechanism of the peninsular effect in passerine birds from South Korea

PLOS ONE

Dear Dr. Hong,

Thank you for submitting your manuscript to PLOS ONE. After careful consideration, we feel that it has merit but does not fully meet PLOS ONE’s publication criteria as it currently stands. Therefore, we invite you to submit a revised version of the manuscript that addresses the points raised during the review process.

We look forward to receiving your revised manuscript.

Kind regards,

Bi-Song Yue, Ph.D

Academic Editor

PLOS ONE

3. We note that Figures 2, 3, 4 in your submission contain map images which may be copyrighted. All PLOS content is published under the Creative Commons Attribution License (CC BY 4.0), which means that the manuscript, images, and Supporting Information files will be freely available online, and any third party is permitted to access, download, copy, distribute, and use these materials in any way, even commercially, with proper attribution. For these reasons, we cannot publish previously copyrighted maps or satellite images created using proprietary data, such as Google software (Google Maps, Street View, and Earth). For more information, see our copyright guidelines: http://journals.plos.org/plosone/s/licenses-and-copyright.

3.1.    You may seek permission from the original copyright holder of Figures 2, 3, 4 to publish the content specifically under the CC BY 4.0 license. 

3.2.    If you are unable to obtain permission from the original copyright holder to publish these figures under the CC BY 4.0 license or if the copyright holder’s requirements are incompatible with the CC BY 4.0 license, please either i) remove the figure or ii) supply a replacement figure that complies with the CC BY 4.0 license. Please check copyright information on all replacement figures and update the figure caption with source information. If applicable, please specify in the figure caption text when a figure is similar but not identical to the original image and is therefore for illustrative purposes only.

Reviewers' comments:

Reviewer's Responses to Questions

**Comments to the Author**

1. Is the manuscript technically sound, and do the data support the conclusions?

Reviewer #1: Yes

Reviewer #2: Yes

2. Has the statistical analysis been performed appropriately and rigorously? 

Reviewer #1: Yes

Reviewer #2: Yes

3. Have the authors made all data underlying the findings in their manuscript fully available?

Reviewer #1: No

Reviewer #2: Yes

4. Is the manuscript presented in an intelligible fashion and written in standard English?

Reviewer #1: Yes

Reviewer #2: Yes

5. Review Comments to the Author

Reviewer #1: The data could be uploaded in a table form per cell in a depository, the referred site is in Korean and it is hard for us to see. The results need more explicit description in the text to be able to follow the findings.

Reviewer #2: Kim et al. peninsular effect

I like this paper very much. The peninsular effect is a fascinating one because it tells so much about the causes of species richness. When those peninsulas face southwards (in the northern hemisphere) then the declining species richness towards the tip runs counter to the large scale patterns of increasing diversity at lower latitudes. This is what this paper finds. Of course, one has to factor out the potentially confounding factors.

I think the authors do that very well. I particularly enjoyed their figures that so clearly articulate the different hypotheses that they compare.

We understand that we will not get surveys from the northern part of the peninsula — for “political reasons” as the authors put it. Having been able to look southwards into North Korea from China (and across the border from Russia), I wonder if they authors might look at the diversity in China’s Amur tiger Amur leopard national park for a comparison of its richness. Which species present there are missing from South Korea? Doing this is not a criterion for publication, incidentally, just something the authors might consider.

One minor quibble

The single model of passerine species richness showed the increasing pattern of species richness with increasing latitude in South Korea. Mean temperature induced by latitude gradient negatively affected LAI and forest area, and positively habitat diversity. However, passerine species richness was only influenced by mean temperature and forest area.

Expressions such as “negatively affected” are hard to understand. So, too, are using acronyms, such as LAI. A better way to write this paragraph would be

The single model of passerine species richness showed species richness increased with latitude in South Korea. Mean temperature increased with decreasing latitude, as did habitat diversity but leaf area index and forest area decline. However, mean temperature and forest area only influenced passerine species richness.

Comparable examples occur throughout.

6. PLOS authors have the option to publish the peer review history of their article (what does this mean?). If published, this will include your full peer review and any attached files.

Reviewer #1: No

Reviewer #2: No

---

## [Author Response · Author response to Decision Letter 0]

3 Jan 2021

To editor

[Commments] In your revised cover letter, please address the following prompts:

[Response] Thanks you for your contribution to this journal as an editor. We’ve checked the manuscript formats and added sentences explaining ethical and legal things in our cover letter.

[Commnents] We note that Figures 2, 3, 4 in your submission contain map images which may be copyrighted. All PLOS content is published under the Creative Commons Attribution License (CC BY 4.0), which means that the manuscript, images, and Supporting Information files will be freely available online, and any third party is permitted to access, download, copy, distribute, and use these materials in any way, even commercially, with proper attribution. For these reasons, we cannot publish previously copyrighted maps or satellite images created using proprietary data, such as Google software (Google Maps, Street View, and Earth). For more information, see our copyright guidelines: http://journals.plos.org/plosone/s/licenses-and-copyright.

[Response] All of these figures (2,3,4) were the ones that we drew by ourselves using the original source data. For example, we obtained bird observation data from the data provider (website: http://ecobank.nie.re.kr), and then we drew the bird richness map (fig 4). Survey region map(fig2) and land cover map(fig3) were also drawn by ourselves.

To reviewer #1 

[Comments] Have the authors made all data underlying the findings in their manuscript fully available? No

[Response] We made and added more information of our results in supplementary 2 and 3 files. And our raw data was uploaded when we upload revision files (raw data, excel file).

[Comments] The data could be uploaded in a table form per cell in a depository, the referred site is in Korean and it is hard for us to see. The results need more explicit description in the text to be able to follow the findings.

[Response] Thanks for your contribution to this paper for reviewer. More information about results of fig.4 and 5 were added in supplementary 2 and 3 files. These results were uploaded in a table form as you recommend (Table S2 and S3). And our raw data was uploaded when we upload revision files (raw data, excel file). And we will ask to make it possible in English for chief of the department proving the web site. Also, we modified parts of abstract and conclusion to make the smooth flow between results and conclusion (line no. 28~44, 291~309).

To reviewer #2

[Comments] I like this paper very much. The peninsular effect is a fascinating one because it tells so much about the causes of species richness. When those peninsulas face southwards (in the northern hemisphere) then the declining species richness towards the tip runs counter to the large scale patterns of increasing diversity at lower latitudes. This is what this paper finds. Of course, one has to factor out the potentially confounding factors.

[Response] Thanks for your contribution to this paper for reviewer. We highlighted our finding in abstract and conclusion section (line no. 38~40, 304~306) as you recommended.

[Comments] I think the authors do that very well. I particularly enjoyed their figures that so clearly articulate the different hypotheses that they compare. We understand that we will not get surveys from the northern part of the peninsula — for “political reasons” as the authors put it. Having been able to look southwards into North Korea from China (and across the border from Russia), I wonder if they authors might look at the diversity in China’s Amur tiger Amur leopard national park for a comparison of its richness. Which species present there are missing from South Korea? Doing this is not a criterion for publication, incidentally, just something the authors might consider.

[Response] As far as we know, Amur tiger and Amur leopard were presumed extinct in the Korean peninsula in early 20th century. We asked a researcher who got interesting in amur tiger and leopard. She said that there is no any record about distribution data in South Korea.

[Comments] One minor quibble

The single model of passerine species richness showed the increasing pattern of species richness with increasing latitude in South Korea. Mean temperature induced by latitude gradient negatively affected LAI and forest area, and positively habitat diversity. However, passerine species richness was only influenced by mean temperature and forest area.

Expressions such as “negatively affected” are hard to understand. So, too, are using acronyms, such as LAI. A better way to write this paragraph would be. The single model of passerine species richness showed species richness increased with latitude in South Korea. Mean temperature increased with decreasing latitude, as did habitat diversity but leaf area index and forest area decline. However, mean temperature and forest area only influenced passerine species richness. Comparable examples occur throughout.

[Response] Thank you so much for your considerate review. We remedied paragraph in abstract and conclusion section (line no. 28~44, 291~309), as you mentioned above. And we refrained using acronyms such as LAI in abstract and conclusion section. And for a detailed understanding of Fig.4 and 5, supplementary 2 and 3 files have been added.

---

## [Editor Report · Decision Letter 1]

12 Jan 2021

Testing the causal mechanism of the peninsular effect in passerine birds from South Korea

PONE-D-20-29245R1

Dear Dr. Hong,

We’re pleased to inform you that your manuscript has been judged scientifically suitable for publication and will be formally accepted for publication once it meets all outstanding technical requirements.

Kind regards,

Bi-Song Yue, Ph.D

Academic Editor

PLOS ONE

---

## [Editor Report · Acceptance letter]

20 Jan 2021

PONE-D-20-29245R1 

Testing the causal mechanism of the peninsular effect in passerine birds from South Korea 

Dear Dr. Hong:

I'm pleased to inform you that your manuscript has been deemed suitable for publication in PLOS ONE. Congratulations! Your manuscript is now with our production department. 

Kind regards, 

on behalf of

Dr. Bi-Song Yue 

Academic Editor

PLOS ONE